# Inhibiting Monocyte Recruitment to Prevent the Pro-Tumoral Activity of Tumor-Associated Macrophages in Chondrosarcoma

**DOI:** 10.3390/cells9041062

**Published:** 2020-04-24

**Authors:** Michele Minopoli, Sabrina Sarno, Gioconda Di Carluccio, Rosa Azzaro, Susan Costantini, Flavio Fazioli, Michele Gallo, Gaetano Apice, Lucia Cannella, Domenica Rea, Maria Patrizia Stoppelli, Diana Boraschi, Alfredo Budillon, Katia Scotlandi, Annarosaria De Chiara, Maria Vincenza Carriero

**Affiliations:** 1Neoplastic Progression Unit, Istituto Nazionale Tumori IRCCS ‘Fondazione G. Pascale’, Naples 80131, Italy; m.minopoli@istitutotumori.na.it (M.M.); g.dicarluccio@istitutotumori.na.it (G.D.C.); 2Department of Experimental Medicine, University of Campania “Luigi Vanvitelli”, 80138 Naples, Italy; sabrina.sarno@unicampania.it; 3Transfusion Medicine Unit, Istituto Nazionale Tumori IRCCS ‘Fondazione G. Pascale’, 80131 Naples, Italy; r.azzaro@istitutotumori.na.it; 4Experimental Pharmacology Unit, Istituto Nazionale Tumori IRCCS ‘Fondazione G. Pascale’, 80131 Naples, Italy; s.costantini@istitutotumori.na.it (S.C.); a.budillon@istitutotumori.na.it (A.B.); 5Division of Musculoskeletal Surgery, Istituto Nazionale Tumori IRCCS ‘Fondazione G. Pascale’, 80131 Naples, Italy; f.fazioli@istitutotumori.na.it (F.F.); m.gallo@istitutotumori.na.it (M.G.); 6Division of Medical Oncology, Istituto Nazionale Tumori IRCCS ‘Fondazione G. Pascale’, 80131 Naples, Italy; g.apice@istitutotumori.na.it (G.A.); l.cannella@istitutotumori.na.it (L.C.); 7Animal Facility, Istituto Nazionale Tumori IRCCS ‘Fondazione G. Pascale’, 80131 Naples, Italy; d.rea@istitutotumori.na.it; 8Institute of Genetics and Biophysics, National Research Council, 80131 Naples, Italy; mpatrizia.stoppelli@igb.cnr.it; 9Institute of Biochemistry and Cell Biology, National Research Council, 80131 Naples, Italy; diana.boraschi@ibbc.cnr.it; 10Laboratory of Experimental Oncology, IRCCS Istituto Ortopedico Rizzoli, 40136 Bologna, Italy; katia.scotlandi@ior.it; 11Pathology Unit, Istituto Nazionale Tumori IRCCS ‘Fondazione G. Pascale’, 80131 Naples, Italy; a.dechiara@istitutotumori.na.it

**Keywords:** chondrosarcoma, monocytes, tumor-associated macrophages, cell migration inhibitors

## Abstract

Chondrosarcomas (CHS) are malignant cartilaginous neoplasms with diverse morphological features, characterized by resistance to chemo- and radiation therapies. In this study, we investigated the role of tumor-associated macrophages (TAM)s in tumor tissues from CHS patients by immunohistochemistry. Three-dimensional organotypic co-cultures were set up in order to evaluate the contribution of primary human CHS cells in driving an M2-like phenotype in monocyte-derived primary macrophages, and the capability of macrophages to promote growth and/or invasiveness of CHS cells. Finally, with an in vivo model of primary CHS cells engrafted in nude mice, we tested the ability of a potent peptide inhibitor of cell migration (Ac-d-Tyr-d-Arg-Aib-d-Arg-NH_2_, denoted RI-3) to reduce recruitment and infiltration of monocytes into CHS neoplastic lesions. We found a significant correlation between alternatively activated M2 macrophages and intratumor microvessel density in both conventional and dedifferentiated CHS human tissues, suggesting a link between TAM abundance and vascularization in CHS. In 3D and non-contact cu-culture models, soluble factors produced by CHS induced a M2-like phenotype in macrophages that, in turn, increased motility, invasion and matrix spreading of CHS cells. Finally, we present evidence that RI-3 successfully prevent both recruitment and infiltration of monocytes into CHS tissues, in nude mice.

## 1. Introduction

Chondrosarcoma (CHS) is the second most frequent diagnosed bone sarcoma after osteosarcoma [1]. CHSs constitute a heterogeneous group of chemo- and radiation-resistant malignant tumors characterized by the production of cartilage matrix [1]. The majority of these patients have a good prognosis after complete surgical resection, as these tumors grow slowly and rarely metastasize, but patients with inoperable disease, due to tumor location, size or metastases, represent a huge clinical challenge to date [2,3]. Histologically, CHSs include conventional, dedifferentiated, mesenchymal, and the chondrosarcoma rare clear cells. Conventional CHS is the most frequent subtype found in 85% of the cases, with a survival rate depending on the histological grade. Among the other less common CHS variants, the dedifferentiated chondrosarcomas (DD-CHS) exhibit a poor prognosis and represent a significant challenge in clinical management, mostly due the co-existence of a high-grade not cartilaginous sarcoma in the context of a low-grade chondrogenic component [3,4]. Despite the occurrence of genetic alterations which have been described in distinct CHS subtypes [5,6,7,8,9], no FDA approved targeted therapies are currently available for CHS [10]. Therefore, the identification of new predictors of tumor progression as well as new treatment options is urgently needed, especially for patients with inoperable or metastatic disease. 

In the last years, the emerging relevance of tumor microenvironment (TME) in cancer progression has led to a shift from a tumor-centered view of cancer development to the concept of a complex tumor ecosystem, in which the cellular and molecular components of microenvironment support multiple aspects of tumor progression [11]. Cancer cells secrete cytokines and chemokines that recruit circulating monocytes from blood into the neoplastic lesions. Subsequently, a plethora of signaling molecules, transcription factors, epigenetic mechanisms, and post-transcriptional regulators contribute to the differentiation of infiltrating monocytes, leading to tumor-educated macrophages with immune-suppressive and pro-tumoral properties [12,13,14]. These tumor-associated macrophages (TAM)s constitute a large portion of the tumor mass, exhibiting predominantly a M2-like pro-tumor phenotype and promoting multiple aspects of cancer progression and metastasis by supporting matrix remodeling, tumor-associated angiogenesis and immune surveillance in a variety of solid tumors [15,16,17,18]. To date, little is known regarding the TME of CHS and its involvement in tumor progression. In this regard, a thorough investigation of the complex crosstalk occurring between CHS cells and TAMs could provide new therapeutic strategies for counteracting CHS progression.

In the past years, we generated the synthetic peptide Ac-d-Tyr-d-Arg-Aib-d-Arg-NH_2_ (named RI-3) that behaves as a potent inhibitor of cell adhesion, migration and angiogenesis [19,20,21].

The RI-3 peptide inhibits the interaction of the Urokinase Receptor (uPAR) with the Formyl Peptide Receptor type 1 (FPR1), both regulators of cell migration. Mechanistically, RI-3 keeps FPR1 anchored to the cell membrane, making it unable to internalize and activate uPAR-triggered, FPR1-mediated cell migration [22]. Both uPAR and FPR1 are expressed in myelomonocytic cells. While uPAR increases with differentiation from monocytes to macrophages [23], contributing to their activation and mobilization, the role of FPR1 is limited to the control of chemotaxis [24,25,26,27,28]. 

In this study, we have analyzed TAM infiltrates in human conventional and dedifferentiated CHS tissues to characterize the functional activity of TAMs in CHS. Then, we have investigated the contribution of CHS cells in triggering a M2-like phenotype and the capability of macrophages to promote growth and/or invasiveness of CHS cells, using non-contact and 3D-organotypic co-cultures. Finally, we investigated whether the novel anti-migratory RI-3 peptide may prevent the recruitment of monocytes into CHS neoplastic lesions engrafted in nude mice. 

## 2. Materials and Methods

### 2.1. Chondrosarcoma Patients

Eighteen patients affected by chondrosarcoma (CHS) were recruited by the Istituto Nazionale Tumori IRCCS “Fondazione G. Pascale” (National Cancer Institute of Naples) in the last ten years. All patients provided written informed consent for the use of tissue samples according to the institutional regulations and the approval by the Ethics Committee of the National Cancer Institute of Naples. Histopathological diagnoses were reviewed on standard H&E-stained slides according to the 2013 WHO classification criteria [1] Medical records were reviewed for clinical information, including the progression-free survival (PFS) whenever possible.

### 2.2. Immunohistochemistry

Immunohistochemistry (IHC) was performed on slides from formalin-fixed, paraffin-embedded tissues, using an automated slide stainer BenchMark (Ventana Medical System-Roche, Monza, Italy). We evaluated the expression of CD68, and CD163 (recognizing all macrophages and macrophages with an M-2 like phenotype, respectively [29]), and CD31, targeting the platelet endothelial cell adhesion molecule-1 on endothelial membranes [30]. Paraffin slides were deparaffinized in xylene and rehydrated through graded alcohols. After antigen retrieval performed according to manufacturer’s instructions, blocking of endogenous peroxidase and unspecific stains were obtained by applying 3% H_2_O_2_ and 1% BSA, respectively, at room temperature for 30 min. Then, slides were exposed to the following primary antibodies: anti-human CD68 clone KP-1 (# 790-2931, ready to use, Roche, Monza, Italy), 1:75 anti-human CD163, clone 10D6, (#NCL-CD163, Leica-Novocastra, Milan, Italy), anti-human CD31, clone JC70A (# M0823, ready to use, Dako, Milan, Italy) for 15 min at 25 °C. For murine tumor tissues, M2 macrophages were visualized with 1:200 diluted anti-mouse MSR1 CD204 antibody (P#MA5-29733, Thermo Fisher Scientific, Milan, Italy). The BOND Polymer Refine Detection (Leica) was used according to manufacturer’s instructions. Finally, sections were counterstained with hematoxylin and mounted. All slides were recorded by a light microscope connected to a video camera and analyzed by using the Axiovision 4.4 software (Carl Zeiss, Milan, Italy). Quantitative evaluation of macrophage and microvessel staining was conducted by two independent pathologists, blinded to clinical information. Sections were scored based on the average counts of positive cells (CD68 and CD163) or microvessels (CD31) counted in the tumor areas, in five randomly selected fields/sample (~6.9 mm^2^) at 200× magnification. For each section, positive cells or microvessels were scored as 1 to 5, according to the following number of positive cells or microvessels encountered in each field: 1 (1–25), 2 (26–50), 3 (51–100), 4 (101–150), and 5 (>150).

### 2.3. Primary Cell Culture

Representative samples from the tumor excision (~1 cm × 1 cm) of patients #8 and #16were immediately minced with a scalpel under sterile conditions and incubated with 1.0 mg/mL collagenase XI (Sigma-Aldrich, Milan, Italy) for 3 h at 37 °C under gentle agitation, as previously described [31]. Cells, recovered by centrifugation at 1500 rpm, were cultured in 6-well multi-dish plates in Dulbecco Modified Essential Medium (DMEM) with the addition of 10% fetal bovine serum (FBS), penicillin (100 U/mL) and streptomycin (100 μg/mL). Isolated cell clusters were further amplified in growth medium until an adherent, homogeneous cell population was obtained. Primary CHS cells stably expressing Green Fluorescent Protein (GFP), were obtained using pEGFP-N1 vector (Clontech, Mountain View, CA, USA) and polyfectamine transfection reagent (Qiagen, Milan, Italy) as described [19]. G418-resistant cells expressing the highest levels of GFP were isolated and amplified. 

### 2.4. Cell Lines

The human monocytic leukemia THP-1 cell line (purchased from the American Type Culture Collection, Manassas, VA, USA) was cultured in RPMI-1640 medium, supplemented with 10% heat-inactivated FBS, penicillin (100 U/mL) and streptomycin (100 μg/mL). THP-1 cells were differentiated into M2 polarized macrophage-like cells by 48 h incubation with 150 nM phorbol 12-myristate 13-acetate (PMA; P8139 Sigma-Aldrich) followed by 48 h incubation with 20 ng/mL interleukin 4 (IL-4; #204-IL, R&D Systems,). Human umbilical vein endothelial cells (HUVEC; C2519A, Lot# 0000115425, with lot-specific certificate of analysis, Lonza Bioscience, Braine-l’Alleud, Belgium) were grown in Eagle Basal Medium (EBM) supplemented with 4% FBS, 0.1% gentamicin,1 μg/mL hydrocortisone, 10 μg/mL epidermal growth factor and 12 μg/mL bovine brain extract (Cambrex, Milan, Italy) [32]. All cells were maintained in atmosphere of humidified air with 5% CO_2_ at 37 °C.

### 2.5. Isolation of Blood Monocytes

Buffy coats were obtained from healthy blood donors at Transfusion Medicine of the National Cancer Institute of Naples, after informed written consent. Human peripheral blood mononuclear cells were harvested by density gradient centrifugation of 50 mL buffy coats mixed with an equal volume of PBS using Lympholyte-poly Cell Separation Media (Cedarlane Laboratories, Milan, Italy). Serum and peripheral blood mononuclear cells (PBMC) were individually collected. PBMCs were washed twice with PBS, counted using the trypan blue dye exclusion method, and monocytes isolated by positive selection of CD14^+^ cells with the Monocyte Isolation Kit II (Miltenyi Biotec Bologna, Italy). The obtained monocytes (88% pure by visual and cytofluorimetric analysis) were transferred to tissue culture plates in RPMI-1640 medium, supplemented with 10% autologous human serum, penicillin (100 U/mL) and streptomycin (100 μg/mL). To obtain murine monocytes, blood samples (about 500 μL/mouse) from the retro-orbital venous plexus of mice anesthetized with 1% isoflurane, were collected using a heparinized capillary tube. PBMCs purified from whole blood with the OptiPrep^TM^ gradient solution (Sigma-Aldrich), were transferred to tissue culture plates in RPMI-640 medium supplemented with 10% heat-inactivated FBS for three days to isolate adherent monocytes/macrophages. 

### 2.6. Collection of Conditioned Media

Primary CHS cells were grown to 80% confluence. Growth medium was removed and cells were extensively washed with PBS and incubated in serum-free medium. To obtain conditioned media (CM) from THP-1 or human monocytes, cells were gently scraped with a plastic cell scraper, recovered by centrifugations at 1100 rpm, washed with PBS and resuspended (0.5 × 10^6^ cells/mL) in serum-free RPMI 1640 medium. After 18 h, CM were centrifuged twice at 2200 rpm for 5 min at 4 °C, and aliquots stored at −80 °C.

### 2.7. 3D Organotypic Cultures

Organotypic co-cultures were carried out as previously described [20,33]. Briefly, 1 × 10^5^ normal, human dermal fibroblasts (NHDF) were starved in serum free medium for 18 h, suspended in 250 μL heat-inactivated serum and embedded in 250 μL α-Minimum Essential Medium 10× containing 2 mg/mL Type I Collagen (#124–25; Cell Application Inc. San Diego, CA, USA), plus/minus 5 × 10^4^ THP1 cells or 1 × 10^5^ human monocytes. Spheroids containing GFP-tagged CHS cells were obtained by using the Perfecta 3D Hanging Drop Plate (3D Biomatrix, Piacenza, Italy). Single GFP-tagged CHS cell suspension (5 × 10^3^ cells in 40 µL/well) was pipetted onto the 3D Hanging Drop Plate lid and left to form spheroids for 72 h, prior to embed them in the collagen/fibroblast mixture, in the presence or in the absence of monocytes. Collagen/fibroblast matrix was allowed to contract until it fitted in a well of a 24-well dish (~5 days), changing growth medium with/without 10 nM RI3 every other day. Images acquired with an inverted fluorescent microscope at 50× magnification allowed us to monitor CHS-spheroid growth in a 7day time frame. Measurement of spheroid size was performed by using the following Equation (1): (1)$$V=\frac{D\left(d^{2}\right)}{2}$$
where D and d are the major and the minor diameter, respectively. 

### 2.8. Non-Contact Co-Culture Assays

THP-1 cells (5 × 10^5^ cells/well) or human monocytes (1 × 10^6^ cells/well) were seeded in the lower compartment of 24 transwell polyethylene terephthalate permeable supports, allowing the exchange of soluble factors (Corning, Milan, Italy) and let to adhere in RPMI-1640 medium supplemented with 10% heat inactivated serum. CHS cells (2 × 10^5^ cells/well) suspended in growth medium, were seeded on the filter top and incubated at 37 °C with 5% CO_2_ for 72 h, changing medium every other day. Then, monocytes were recovered and their phenotype analyzed by a flow cytometer. After co-culture, CHS were removed and CM from THP-1 and human monocytes prepared as described above, were analyzed for cytokines chemokines and growth factors by dot blot or bio-plex immunoassays. The experiments were performed three times.

### 2.9. Peptide Synthesis

The peptide RI-3 (Ac-d-Tyr-d-Arg-Aib-d-Arg-NH_2_) was custom-synthesized on solid-phase with Fmoc/t-Bu chemistry by JPT Peptide Technologies GmbH (Berlin, Germany). RI-3 was purified by reversed-phase HPLC using water/acetonitrile gradients, and characterized by UPLC-MS [34].

### 2.10. Cytofluorimetric Analysis

To analyze changes in monocytic phenotype, cells were exposed to the following fluorochrome conjugated antibodies: PE-conjugated anti-CD14, APC-Cy7-conjugated anti-CD68, APC-conjugated anti-CD163, PE-Cy7-conjugated anti-CD206 (all from Miltenyi Biotec, Bologna, Italy). Fluorochrome-conjugated, isotype-matched control antibodies were included to assess background fluorescence. All incubations were carried out in 0.5% bovine serum albumin, supplemented with the corresponding immunoglobulin G to minimize nonspecific binding. Samples were acquired with the BD FACSCanto II (BD Biosciences, Milan, Italy), and data analyzed by the FlowJo v10.0.7 software (Tree Star, Inc. Ashland, OR, USA), after gating on the myeloid population in the FSC/SSC plot. Values were expressed as the percentage of each specific marker over median fluorescence intensity of the unstained cells.

### 2.11. Dot Blot Array

The relative levels of soluble factors secreted by THP-1 cells after co-cultures with primary CHS cells, were analyzed using the dot blot Human Cytokine Array Kit panel A (#ARY005B, R&D Systems, Milan Italy), according to the manufacturer’s instructions. Briefly, 500 mL CM were applied on each membrane, and signals were detected using the streptavidin-horseradish peroxidase and chemoluminescent detection reagents. The pixel density of each spot was measured using the NIH Image J 2.0 software. Positive control spots were utilized to normalize results between the membranes. The intensity for each spot was then averaged over the duplicate spots. The experiment was performed three times.

### 2.12. Bio-Plex Assay

To evaluate the cytokine, chemokine and growth factor levels in the monocytes CM, we used the Bio-Plex Pro Human Cytokine 27-Plex Immunoassay. Protein levels were determined using a Bio-Plex array reader (Luminex-Bio-Rad Laboratories, Milan, Italy). The levels of cytokines were measured using a standard curve, generated by the software provided by the manufacturer (Bio-Plex Manager Software, version 4.0, Luminex). The experiment was performed twice in quadruplicate.

### 2.13. Cell Proliferation

Cell proliferation was assessed using E-16-well plates and the xCELLigence Real Time Cell Analysis (RTCA) technology (Acea Bioscience-CaRli biotec, Rome, Italy) as described [20]. Briefly, CHS cells (2 × 10^3^/well) were seeded in 16-well E-plates in CM recovered from human monocytes-CHS co-cultures, or in CM from CHS cells alone, the last as control. All CM were supplemented with 5% heath-inactivated FBS, and cells were grown for 96 h. Microelectrodes placed on the bottom of plates, detect impedance changes, which are proportional to the number of adherent cells and are expressed as Cell Index. The impedance value of each well was automatically monitored by the xCELLigence technology (CaRli biotec, Rome, Italy) and expressed as a Cell Index value. Doubling times were calculated from the cell growth curve during the exponential growth. The experiment was performed twice in quadruplicate.

### 2.14. Cell Migration and Invasion in Boyden Chambers

Chemotaxis assays were performed in Boyden chambers, using 8 μm pore size PVPF-filters (Nucleopore-Merk, Darmstadt, Germany) as previously described [35]. Briefly, 1 × 10^4^ viable CHS cells were seeded in each upper chamber in serum-free medium. The lower chamber was filled with medium (CTRL), CM from THP-1 cells recovered after THP-1/CHS co-cultures or CM from CHS cells alone. In all cases, serum was added to a 5% final concentration in the lower compartment of Boyden chambers. Cells were allowed to migrate for 4 h at 37 °C, 5% CO_2_. For the invasion assays, filters were coated with 50 μg/filter matrigel (BD Biosciences, Milan, Italy) and cells (3 × 10^4^ viable cells/well) were allowed to invade matrigel for 18 h at 37 °C, 5% CO_2_. In all cases, at the end of the assay, cells on the lower filter surface were fixed with ethanol, stained with hematoxylin and 10 random fields/filter were counted at 200× magnification. The arbitrary value of 100% was given to the basal cell migration or invasion assessed in the absence of chemoattractant. All experiments were performed three times in triplicate, and the results expressed as percentage of the basal cell migration or invasion.

### 2.15. Trans-Endothelial Migration

Trans-endothelial migration assays performed using the xCELLigence RTCA technology as described [19]. Briefly, HUVEC (2 × 10^4^ cells/well) were suspended in growth medium, plated on E-16-well plates and grown for ~25 h until they formed a confluent monolayer. Then, THP-1 cells, human or murine monocytes (5 × 10^4^ cells/well) were seeded in growth medium in the presence or the absence of 10 nM RI-3. When HUVECs are challenged with crossing cells, there is a drop in electrical resistance which is monitored in real-time for 10 h as the cell index changes (due to crossing of the endothelial monolayer). The experiment was performed twice in quadruplicate.

### 2.16. In Vivo Experiment

To evaluate the effect of RI-3 on intratumoral monocyte infiltration, CHS cells were injected subcutaneously, as a single-cell suspension (1 × 10^6^ cells in 100 µL sterile PBS, 97% viability), in the right flanks of ten six-eight week old, Foxn1^nu/nu^ female nude mice (Harlan, San Pietro al Natisone, Italy) of 22 to 25 g. Animals were randomized into two 5-mice groups with the treatment group receiving 6 mg/kg RI-3 by intra-peritoneal injection every 24 h, and the control group receiving an equivalent injected volume of vehicle (PBS) as described [19]. After 12 days, the animals were sacrificed, the excised tumors fixed in buffered formalin and processed for paraffin sectioning. CD204^+^ cells revealed by IHC were counted in 5 randomly chosen fields per section, in at least two sections/tumor at 200× magnification. 

### 2.17. Statistical Analysis

Data are expressed as the means ± SD of the number of the indicated determinations. Data derived from in vitro experiments were analyzed by one-way ANOVA post hoc Dunnett *t*-test for multiple comparisons. *p* < 0.05 was accepted as significant. Pearson’s correlation test was employed to analyze the correlations between CD68, CD163, and CD31 expression, histology and clinicopathologic parameters, assessed by using the SPSS 20.0 software (SPSS Inc. Chicago, IL, USA).

### 2.18. Ethics Statement

All experimental protocols were performed in accordance with guidelines of the Istituto Nazionale Tumori “Fondazione G. Pascale”-IRCCS (Quality System n. LRC 6019486/QMS/U/IT- 2015 certificated in conformity with UNI EN ISO 9001:2008). The research work with primary cell lines and CHS tissues has been approved by Institutional Ethical Committee of Istituto Nazionale Tumori “Fondazione G. Pascale”-IRCCS, Naples, Italy (protocol 258/18, December 2018). The care and use of animals were approved by Institutional Ethical Committee of Istituto Nazionale Tumori “Fondazione G. Pascale”-IRCCS, Naples, Italy and by the Italian Ministry of Health (protocol n.1185/2016-PR).

### 2.19. Data Availability

All data generated during this study are available within the article and its Appendix A. Further details are available from the corresponding author on reasonable request.

## 3. Results

### 3.1. Density and Distribution Patterns of Macrophage Infiltration and Microvessels in CHS Tissues 

Monocyte-derived macrophages are recruited and reprogrammed by tumor cells (tumor-associated macrophages or TAMs) and have been documented to promote angiogenesis in several types of solid tumors [13,36]. Furthermore, it has been documented that microvascularity associates with an aggressive clinical behavior and a high metastatic potential in chondrosarcomas (CHS) [37,38]. Therefore, we investigated the relationship between TAMs, intratumor vascularization and aggressiveness in CHS. To characterize the chondrosarcoma-associated macrophages, tissue samples from 18 patients having a median age of 60 years (range, 34–79 years), whose clinicopathological characteristics are summarized in Table 1, were analyzed. None of the patients received neoadjuvant chemotherapy or radiotherapy before undergoing surgical resection. The median tumor size was 12 cm (range, 4–22 cm). All specimens were from the resection of the primary tumor and include 6 dedifferentiated CHS (DD-CHS) and 12 conventional CHS. Conventional CHS were graded as G1, G2 or G3 according to 2013 WHO Classification (Table 1). Progression free survival (PFS) was calculated by reviewing the medical records of only eight patients enrolled between 2009 and 2015, the others being accrued between 2016 and 2019. Metastatic lesions occurred in five CHS (#1, 2, 3, 4, 10) and in two DD-CHS (#14, 16) patients. CHS patients #2, 3 and 4 died a few months after surgery. 

In this cohort of patients, we examined the localization and the abundance of CD68^+^ macrophages, CD163^+^ TAM and CD31^+^ intratumoral microvessels. After immunohistochemistry (IHC) carried out on FFPE tissue sections, all CHS tissues appeared infiltrated by CD68^+^ macrophages, although to a different extent (Figure 1). For each tissue sample, the average of positive CD68 and CD163 cells as well as microvessels were scored from 1 to 5, depending on the number of positive cells, as reported in Methods. In Figure 1, representative images of tissue samples stained with anti-CD163 from patient #8 (score 5), from patient #16 (score 4), patient #14 (score 3), from patient #13 (score 2) and patient #10 (score 1) along with the corresponding CD68 and CD31 staining are presented. In all CHS tissues, CD68^+^ cells were more numerous than CD163^+^ cells, in absolute terms (Figure 1 and Table 2). We found that, while in DD-CHS tissues macrophages were homogeneously distributed, in conventional CHS tissues, they localized preferentially at the margin of cartilaginous nodules (Figure 1). In both CHSs and DD-CHSs, perivascular areas appeared colonized by clusters of CD68^+^ cells. As assessed by CD31 staining, both CHS and DD-CHS tissues exhibit an appreciable and comparable intratumoral microvessel density (iMVD), which appeared mainly localized at the margin of cartilaginous nodules in conventional CHS tissues (Figure 1, Table 2). 

By using the Pearson’ correlation test, averages and scores of CD68, CD163, and iMVD were subjected to statistical analysis (Appendix A). As expected, a statistically significant correlation was found between averages and scores of CD68^+^ and CD163^+^ cells (Figure 2B and Appendix A). Statistically significant correlation between CD163 and iMVD averages and scores was higher as compared to that occurring between averages and scores of CD68 and iMVD (Figure 2C,D and Appendix A). Notably, CD163^+^ infiltrations do not correlate with age and tumor size. Although no statistical evaluation was applicable for the paucity of cases with ascertained 5-years follow-up (only eight CHS cases), we noted that high CD163 and iMVD averages inversely correlate with PFS (Appendix A), suggesting a link between TAM abundance and poor prognosis in CHS. 

### 3.2. Monocytes Increase Spreading of CHS Spheroids Embedded Into Collagen-Fibroblast Matrices

Accumulating evidence indicates that: i) Cancer cells drive infiltrating monocytes and macrophages toward an M2-like pro-tumoral phenotype [15]; ii) tumor-associated macrophages (TAMs) orchestrate many stages of tumor progression by secretion of proteases, angiogenic substances, growth factors, and cytokines, depending on their activation status [39]. To investigate whether CHS-associated monocyte/macrophages may engage a crosstalk with CHS cells, organotypic co-cultures were set up, reproducing TME in a 3D-environment. In this system, monocytes and CHS spheroids were incorporated in a semi-solid matrix containing dermal fibroblasts. Then, size and spreading of tumor spheroids were monitored for 7 days. 

Primary CHS cells derived from tumor samples of patients #8 and #16 were subjected to enzymatic digestion and amplified until an adherent, homogeneous cell population was obtained (Figure 3A,B). Primary tumor cells were stably transfected with a GFP plasmid and then allowed to form spheroids for 72 h. Spheroids were dropped into collagen matrices combined with dermal fibroblasts with/without THP-1 cells before the starting of contraction. Time-dependent increase of spheroid size was monitored for 7 days by acquiring images with bright field and fluorescence microscopy. Fibroblast-dependent matrix deposition allows spheroid growth in control samples (Figure 3C,D). Remarkably, the inclusion of THP-1 cells into organoids caused a dramatic increase in size and spreading of spheroids derived from both #8 and #16 primary CHS cells (Figure 3C,D).

Measurement of spheroid volumes at day 7 revealed that THP-1 cells cause an about 80% and 60% size increase of #8 and #16 CHS spheroids, respectively (Figure 3E). Like THP-1, monocytes isolated from healthy donors caused a time-dependent increase of spheroid size (Figure 4A,B), confirming the ability of monocytes to promote CHS cell ability to grow and/or infiltrate surrounding tissues.

To understand whether the monocyte-dependent increase in the CHS spheroid size was due to an increased proliferation or to a spreading effect, we sought to determine the rate of proliferation of primary CHS cells using the xCelligence technology. Primary CHS cells were exposed to conditioned medium (CM) of human monocytes co-cultured with CHS cells or CM from CHS control cells. In all cases, heath-inactivated serum was included to a 5% final concentration. CHS exposure to both CM did not affect significantly the resulting proliferation curves, showing very similar doubling times (12.11 h and 11.87 h, respectively), suggesting that pro-invasive factors may be produced as a consequence of the interaction between the two cell types (Appendix A). Indeed, CM recovered after co-culture of monocytes with CHS elicited a 40% increase of CHS cell motility and invasive capability, as compared to CM from CHS alone (Appendix A). Collectively, these findings indicate the existence of a crosstalk between monocytes and CHS cells and suggest that the monocyte-induced increase in spheroid size is mainly due to spreading of CHS into the semi-solid matrix-fibroblasts mixture. 

### 3.3. CHS Cells Can Educate Macrophages Toward the M2-Like Functional Phenotype 

To assess possible changes in monocyte phenotype, following co-cultures with CHS cells, THP-1 cells were co-cultured with primary CHS cells obtained from patient #16, in an in vitro not-contact transwell co-culture assay. Interestingly, after 72 h of co-culture with CHS cells, the majority of THP-1 cells became adherent with rounded and/or spindle-shaped morphology, very similar to that acquired after treatment with PMA/IL-4 (Appendix A). The phenotype of THP-1 cells was analyzed by flow cytometry using CD163 and CD206 to identify the alternatively activated M2 macrophage phenotype [40]. After co-culture with CHS cells, THPI-1 cells, express significantly higher levels of CD68 CD163 and CD206 as compared to control THP-1 cells (Figure 5A,B). 

The expression levels of CD163 and CD206 in THP-1 cells recovered from co-culture with CHS were quite comparable to those assessed in PMA/IL-4 -treated THP-1 (Figure 5B,C and Appendix A), suggesting a tumor-promoted differentiation/polarization towards M2-like macrophages. After co-culture, CHS cells were removed, and CM from THP-1 cells, prepared as described in Materials and Methods, was analyzed by a dot plot assay for the content of cytokines and chemokines characterizing the M2-like phenotype. THP-1 cells co-cultured with CHS cells secreted high levels of IL-10 and negligible levels of IL-12 with respect to control THP-1 (Figure 5D,E), supporting again the notion that CHS cells promote macrophage polarization in the direction of an M2-like pro-tumor functional phenotype [41]. Interestingly, the cross-talk between THP1 and CHS cells caused a dramatic increase of the CC-chemokine ligand 2 (CCL2) (Figure 5D,E), which has been shown to regulate M2-like polarization by downmodulating inflammatory cytokine production [42]. 

To confirm these results with primary human blood monocytes, a similar co-culture system with CHS cells was set up. Monocytes recovered from co-cultures with CHS cells exhibited an appreciable increase in the number of CD68, CD163, and CD206 expressing cells, as compared to control monocytes (Figure 6A–C).

CM from monocytes that had been co-cultured with CHS, contained higher levels of CCL2 and IL-10 and lower levels of IL-12 as compared to CM from control monocytes (Figure 6D). It is known that M2-like-TAMs secrete angiogenic and growth factors, including vascular growth factor (VEGF), basic fibroblast growth factor (bFGF) and platelet-derived growth factor (PDGF) [43]. Accordingly, we found appreciable levels of VEGF, bFGF, and PDGF-ββ in CM from monocyte co-cultured with CHS cells (Figure 6D), further supporting the notion that soluble factors produced by CHS induce a M2-like phenotype in macrophages. 

### 3.4. The RI-3 Peptide Prevents Monocyte Recruitment and CHS Infiltration by TAMs 

Monocytes are considered the primary source of TAMs [44,45,46]. Thus, the inhibition of monocyte mobilization and macrophages accumulation at the tumor site could be considered a valid therapeutic option for CHS. In the past years, we have developed the urokinase receptor (uPAR)-derived peptide, denoted RI-3, which is a potent inhibitor of cell migration [19]. The possibility that RI-3 may effectively counteract the capability of monocyte to cross endothelial monolayers was analyzed by using the xCELLigence RTCA technology, as previously described [47]. HUVEC were allowed to grow until they formed a monolayer for 25 h prior to seeding on top monocytes in the presence of 10% serum, plus/minus RI-3. At this time, reduction of impedance values, due to invading cells that interrupt monolayers was monitored in real-time for 10 hours. As shown in the Figure 7, THP-1 cells, (A), human monocytes (B), and murine monocytes (C) were able to cross the endothelial monolayer, although to a different extent. In all cases, the addition of 10 nM RI-3 inhibited the capability of monocytes to cross the endothelial monolayer, suggesting that RI-3 may prevent monocyte recruitment into CHS tissues.

This hypothesis was investigated in vivo by engrafting primary human CHS (derived for tumor tissue of the patient #16) in nude mice and subsequently administering RI-3 daily for 10 days, according to a previously published procedure [19]. After 12 days, animals were sacrificed, and the tumors were excised, fixed in buffered formalin and processed for paraffin sectioning. M2-like macrophages were identified by IHC as CD204^+^ cells (Figure 7D). We found a statistically significant reduction of CD204^+^ cells in tumors from RI-3 treated mice as compared to those treated with vehicle only (141.2 ± 17 and 91.9 ± 15 CD204^+^ cells/field, respectively) (Figure 7E). 

Taken together, our findings indicate that soluble factors produced by CHS induce an M2-like phenotype in macrophages that, in turn, increase spreading of CHS cells into matrices. In this context, RI-3 peptide successfully prevented both recruitment and infiltration of monocytes into the CHS tumor tissue. 

## 4. Discussion

Chondrosarcomas (CHS) constitute a malignant group of rare cartilaginous matrix-producing neoplasms, with diverse morphological features and clinical behavior [1]. Patients diagnosed with chondrosarcomas (CHS) are subjected to large tumor resections to prevent local recurrences or metastasis, including amputations, with physical disabilities that highly affect daily life. Despite several preclinical studies and clinical trials aimed to identify druggable targets that may improve the prognosis of CHS patients, this bone sarcoma remains an orphan disease, mainly because all studies are performed with small numbers of patients [2,3,4]. Furthermore, no effective treatments exist for advanced CHSs due to their resistance to radiotherapy and chemotherapy. In recent years, the growing interest in cancer immunotherapy reached the sarcoma field and a number of molecular profiling studies led to the identification of immune therapeutic targets in bone sarcomas [48]. PD1 expression seems to have prognostic and therapeutic implications in CHS, whereas PD-L1and T-cell infiltrate were found highly expressed in dedifferentiated CHS [49]. Unfortunately, he clinical responses in trials remain unsatisfactory, suggesting the need for better characterize the CHS microenvironment in an effort to improve the immunotherapeutic response.

Emerging evidence suggests that tumor-associated macrophages (TAM)s promote tumor progression exerting immunosuppressive and pro-angiogenic activities [50,51,52,53,54], In CHS, TAMs constitute the main immune population [55]. 

In this study, a quantitative evaluation of macrophages infiltrating CHS tissues, using CD68 as a macrophage marker and CD163 as a marker of alternatively activated M2 macrophages was conducted [56] and the results were compared with the intra-tumor microvessel density (iMVD). The results show that: (i) a high correlation between CD163^+^ macrophages and iMVD (*r* = 0.929) exist, indicating that a link between TMA abundance and vascularization occur in CHS; (ii) CHS cells trigger an M2-like phenotype in monocyte-derived primary macrophages that, in turn, promote invasiveness of CHS cells, as assessed by 3D-organotypic co-cultures. (iii) recruitment of monocytes into CHS neoplastic lesions engrafted in nude mice may be prevented by the concomitant treatment with the anti-migratory novel RI-3 peptide.

By analyzing the microenvironment in tumor tissues of 26 CHS patients, Simard et al., showed that the number of CD163^+^ cells in tumor tissues positively correlates with the CHS progression [57]. In line with these findings, we observed an inverse correlation between CD163^+^ macrophages and progression free survival (PFS), as well as between high vascularization and PFS, supporting the notion that TAM abundance in CHS tissues adversely affects the prognosis of CHS patients. However, it should be considered that our analysis concerned eighteen CHS cases, due to the low frequency of this tumor and to the lack of a 5-years follow-up for the other recruited patients. Future work will extend this analysis to a larger cohort of patients to evaluate the reliability of TAM abundance as a negative prognostic marker. 

By 3D organotypic co-cultures and non-contact co-cultures, we documented for the first time the occurrence of a crosstalk between CHS cells and monocytes through soluble mediators. Once in the presence of CHS cells, primary blood monocytes induce an increase in CHS spheroid size, mainly due to spreading of CHS cells into the matrix-fibroblasts mix. Regarding polarization status, under these conditions, monocytes acquire a M2-like anti-inflammatory phenotype. Accordingly, conditioned media from monocytes co-cultured with CHS cells contain higher levels of IL-10, bFGF, CCL2, PDGF-ββ, and VEGF, and lower levels of IL-12, as compared with unexposed monocytes. 

In this regard, we foresee that identification of soluble factors secreted by CHS cells in the microenvironment milieu and responsible for M2-like polarization, could allow the development of new-targeted therapies aimed to counteract TAM pro-tumoral functions. Accumulating studies document that CAFs are engaged in a reciprocal relationship with TAMs, that may promote cancer progression through the release of large amounts of ECM proteins, cytokines and soluble factors [58,59,60]. Our recent studies revealed that mammary epithelial cells expressing c-Myc oncogene recruit and activate primary fibroblasts in a paracrine manner through the IGFs/IGF-1R axis, finally promoting matrix invasion by mammary epithelial cells [33]. In this system, modeling the early stages of breast tumor-stroma crosstalk in 3D-organotypic cultures with primary fibroblasts, it would be interesting to study the progressive differentiation steps underlying the acquisition of the M2 phenotype. Another aspect to be investigated is the possible, contribution of CAFs to elicit macrophage polarization, in the context of mesenchymal tumors like CHS. 

It is widely accepted that TAMs originate mostly from circulating precursor monocytes that infiltrate tumor tissues differentiating into macrophages [61]. Due to their intrinsic plasticity, they may shift between an M1-like pro-inflammatory to M2-like anti-inflammatory phenotype polarization status, or specific M2 macrophage subsets, depending on external stimuli [62], suggesting that new therapeutic options could include the manipulation of tumor microenvironment and its immune infiltrates. Indeed, many efforts have been directed to the prevention of monocyte recruitment into tumor tissues, counteracting their M2-like polarization, or, alternatively, forcing their phenotype toward a M1 pro-inflammatory phenotype with new compounds now in preclinical and clinical trials [63,64]. However, some of these strategies have important limitations due to their specificity and toxicity therefore need further investigation [64]. 

We and others have documented that the expression of urokinase receptor (uPAR) in monocytes increases with differentiation to macrophages and contributes to their activation, thus regulating immune responses, inflammation, and tumor progression [24,25]. Also, in a variety of cancer cells, uPAR expression has been shown to induce, secretion of IL-4, via a mechanism that involves activation of ERK1/2, which, in turn, promotes M2 polarization of macrophages [65]. On the other hand, we have already documented that: (i) uPAR is highly expressed in human CHS tissues [31]; (ii) CHS tumor cells expressing uPAR and the formyl-peptide receptor type 1 (FPR1) acquire the ability to migrate and invade basal membranes [19]; (iii) uPAR triggers cell migration through the interaction of its 84–95 sequence with the FPR1 [35].

Previous work from this laboratory has shown that uPAR-derived synthetic peptides carrying the S90P or S90E amino-acidic substitutions, up- or down-regulate cell migration, respectively [35]. Following this observation, we developed a series of linear peptides containing substitution of Ser90 with the glutamic acid or α-aminoisobutyric acid residues in the uPAR sequence that inhibit the uPAR/ FPR1 interaction and reduce to basal levels directional cell migration [32,66]. While providing proof-of-principle for the strategy, none of these peptides represents an ideal lead molecule: some of these peptides are unstable to enzymatic digestion in human serum, whereas others exhibit toxicity when administered in vivo, probably due to a low affinity binding site to the alpha chain of vitronectin receptor [32,66]. To generate more stable uPAR/FPR1 inhibitors, we applied the retro-inverso approach to our previously described uPAR/FPR1 inhibitors [19]. Among these, the retro-inverso peptide Ac-d-Tyr-d-Arg-Aib-d-Arg-NH_2_, named RI-3 is stable RI-3 is stable in human serum and has no effect on cell proliferation, even at a 10 μM concentration [19,20]. At nanomolar concentrations, it inhibits migration, matrigel invasion, and trans-endothelial migration of human sarcoma cells [19]. In nude mice engrafted with CHS cells, RI-3 caused a significant reduction of circulating tumor cells and intra-tumor vascularization, the latter being due to prevention of the VEGF-driven angiogenesis [19].

These findings suggested that RI-3 could exert similar effects on the monocyte recruitment into CHS tissues. Here, evidence is presented that RI-3 inhibits trans-endothelial migration of human and murine blood monocytes. Remarkably, following subcutaneous injection of primary CHS cells in nude mice, a daily treatment with 6 mg/kg RI-3 significantly reduced the number of TAMs counted in tumor sections, as compared to control animals. Although we cannot exclude that the RI-3-dependent decrease in TAM number may also depend on the reduction of intra-tumoral vascularization, these findings indicate that RI-3 successfully prevents both recruitment and infiltration of monocytes into the tumor tissues. Thus, RI-3 may be considered a promising lead for development of new therapeutic strategies aimed to counteract the pro-tumoral effects of TAMs in chondrosarcoma. This information, together with the potent anti-angiogenic activity of RI-3, encourage to consider this peptide as a promising lead compound for development of new therapeutic strategies aimed at counteracting CHS progression.

## Figures and Tables

**Figure 1 cells-09-01062-f001:**
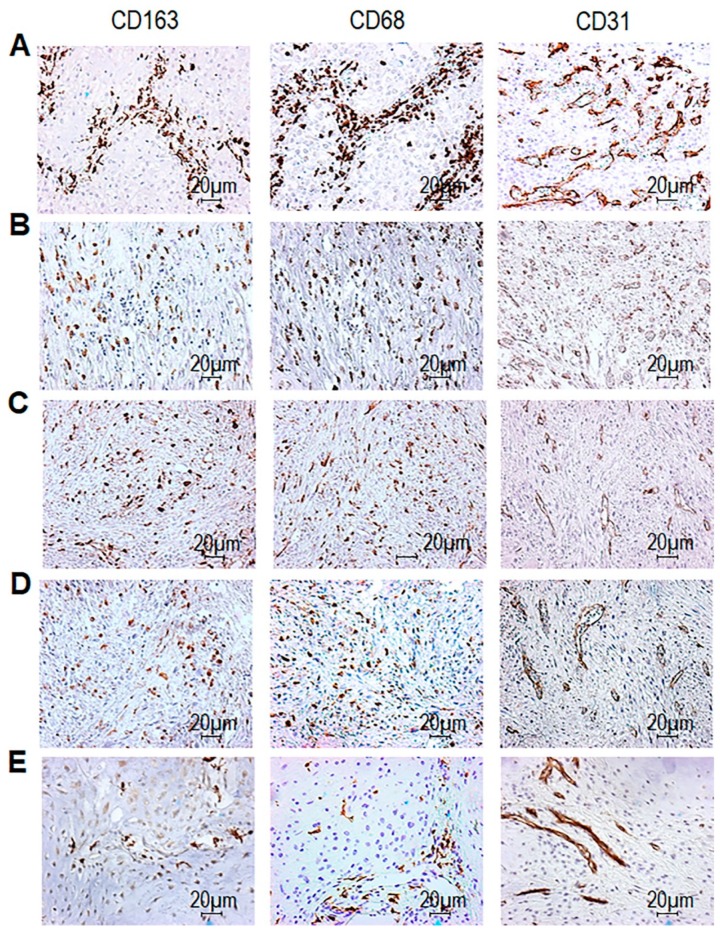
CD68, CD163, and CD31 expression in tumor tissues from chondrosarcomas (CHS) patients. CHS tissues were processed for IHC analysis of CD68^+^ and CD163^+^ cells, and CD31^+^ microvessels. Sections were scored based on the average counts of positive cells (CD68 and CD163) or microvessels (CD31) counted in the tumor areas, in five randomly selected fields/sample at 200× magnification. For each section, positive cells or microvessels were scored as 1-5 where 1 (1–25), 2 (26–50), 3 (51–100), 4 (101–150), and 5 (> 150) positive cells or microvessels were encountered in the field. Representative images of CD163 score 5 (**A**, from patient #8) score 4 (**B**, from patient #16) score 3 (**C**, from patient #14) score 2 (**D**, from patient #13) and score 1 (**E**, from patient #10). The corresponding CD68 and CD31 immunostaining are shown in the central and right columns (scoring is reported in the Table 2). Original magnification: 200×. Distribution of CD68, CD163 and i intratumoral microvessel density (iMVD) scores are reported in Table 2 and summarized in Figure 2A.

**Figure 2 cells-09-01062-f002:**
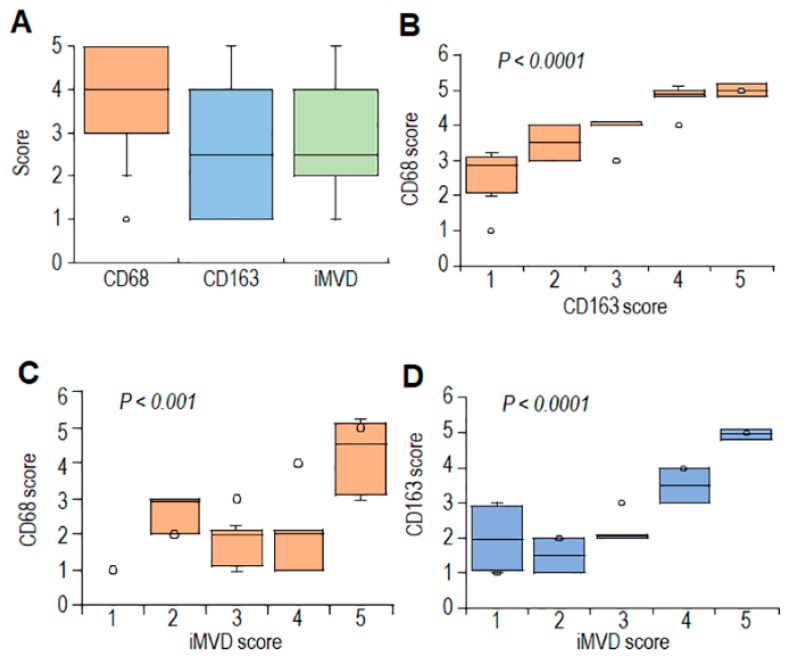
Relationships between CD68, CD163, and CD31 expression levels in tumor tissues from CHS patients. (**A**). Box plot, showing variation in the distribution of CD68, CD163 and intratumor microvessel density (iMVD) scores in CHS tissues. (**B**,**C**). Box plots showing CD68 scores in CHS tissues according to CD163 (**B**) and iMVD (**C**) scores. (**D**). Box plot, showing scores of CD163 scores in CHS tissues according to iMVD scores. Dark horizontal lines represent the medians. Circles represent outliers.

**Figure 3 cells-09-01062-f003:**
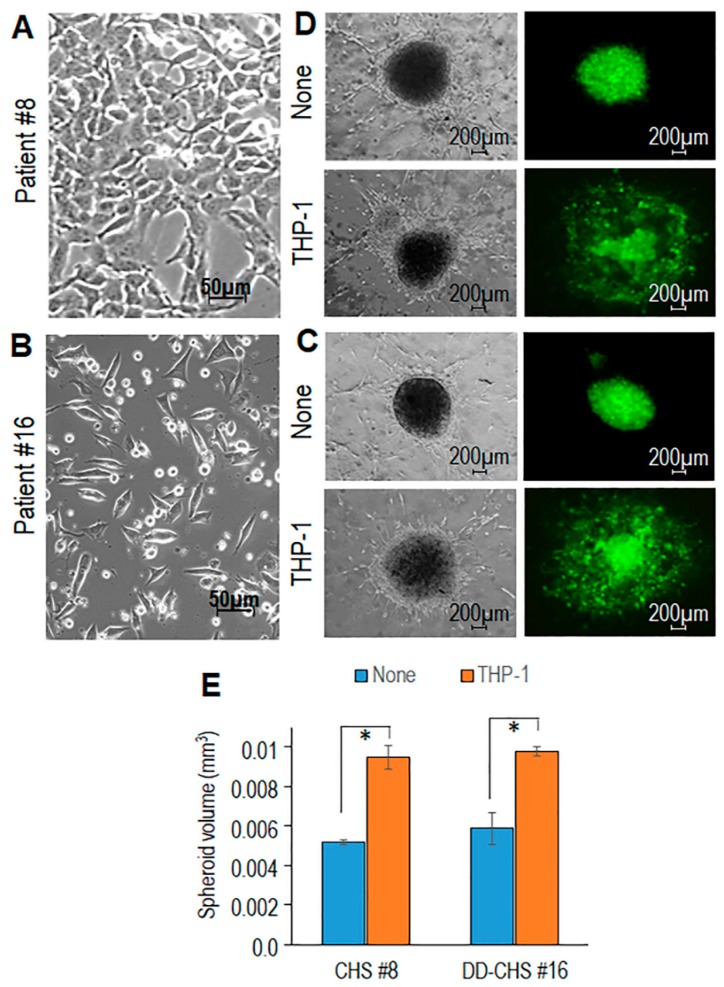
Contribution of THP-1 cells in promoting spreading of primary CHS cells in organotypic co-cultures. Primary CHS cells obtained from the tumor samples of #8 (**A**) and #16 (**B**) patients visualized by phase contrast microscopy. Original magnification: 200x. (**C**,**D**). Spheroids of GFP-tagged CHS cells (**C**: patient #8, **D**: patient #16) were embedded in a collagen/fibroblast mixture, without (None), or with the addition of THP1 cells. Fluorescent and transmitted-light input images were acquired after 7 days at 50× magnification. (**E**). Spheroid sizes assessed after 7 days by using the Equation (1), where D and d are the major and the minor diameter, respectively. Data are the mean ± SD of two independent experiments, performed in duplicate. Statistical significance with * *p*  <  0.0001.

**Figure 4 cells-09-01062-f004:**
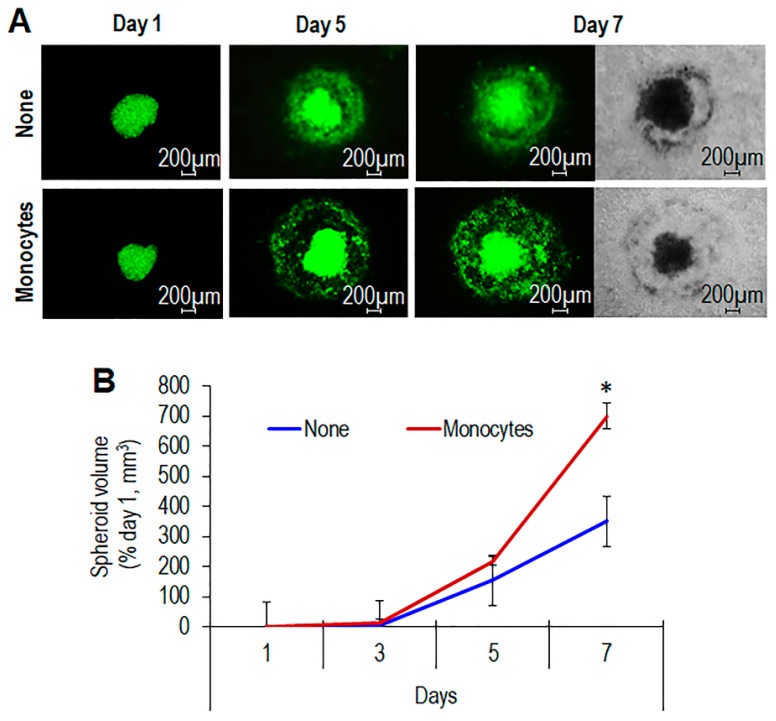
Time-dependent increase of spheroid size induced by primary monocytes. (**A**). Spheroids containing GFP-tagged CHS cells obtained from the tumor sample of #16 patient were embedded in the collagen/fibroblast mixture without (None), or with the addition of human monocytes. At the indicated times, fluorescent and transmitted-light input images were acquired at 50× magnification. (**B**). Time-dependent increase of spheroid size. Data expressed as percentage of volumes assessed at time zero are the mean ± SD of two independent experiments, performed in duplicate. Statistical significance with * *p* < 0.0001.

**Figure 5 cells-09-01062-f005:**
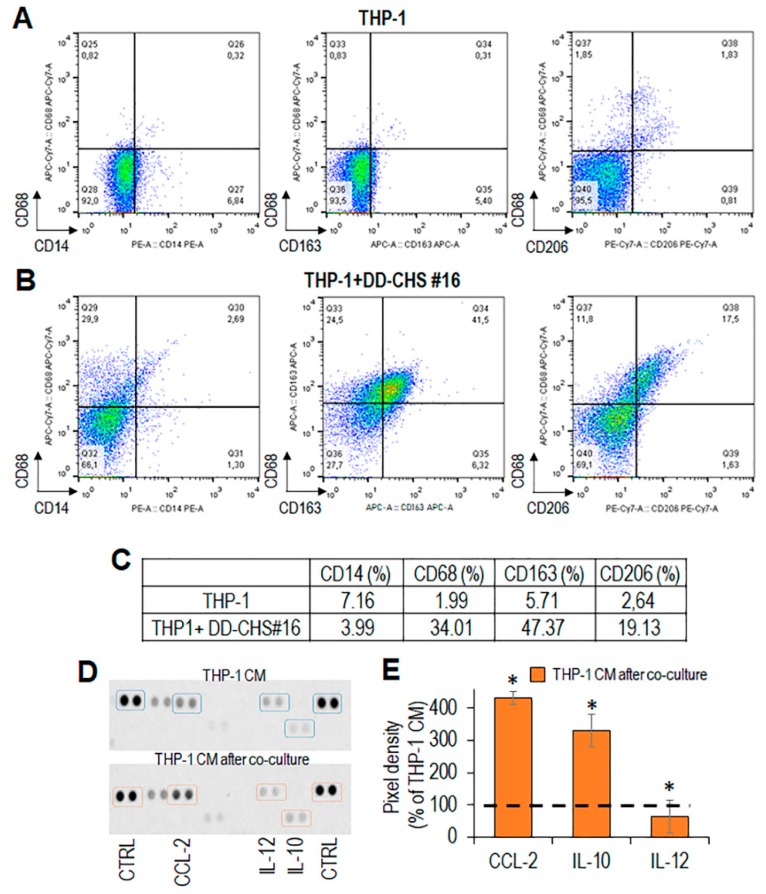
Immunophenotyping of THP-1 cells co-cultured with CHS cells. THP-1 cells were co-cultured with primary CHS cells for 72 h. (**A**,**B**). Phenotypic analysis of THP-1 cells, collected after co-culture, by flow cytometry. (**C**). Percent variation of CD14^+^, CD68^+^, CD163^+^, and CD206^+^ cells upon co-culture with CHS cells, compared to control (THP-1 alone). (**D**). After co-culture, CHS cells were removed, THP-1 conditioned media prepared as described and analyzed for the content of CC2, IL-10 and IL-12 by a dot plot assay. (**E**). The pixel density of each spot was measured using NIH Image J 2.0 software. Positive control spots were used to normalize results between the membranes. The intensity of each spot was averaged over the duplicate spots and expressed as percentage of each cytokine or chemokine spontaneously secreted by THP-1 cells, considered as 100% (dashed line). Data represent mean ± SD from three experiments performed in quadruplicate with * *p* < 0.005.

**Figure 6 cells-09-01062-f006:**
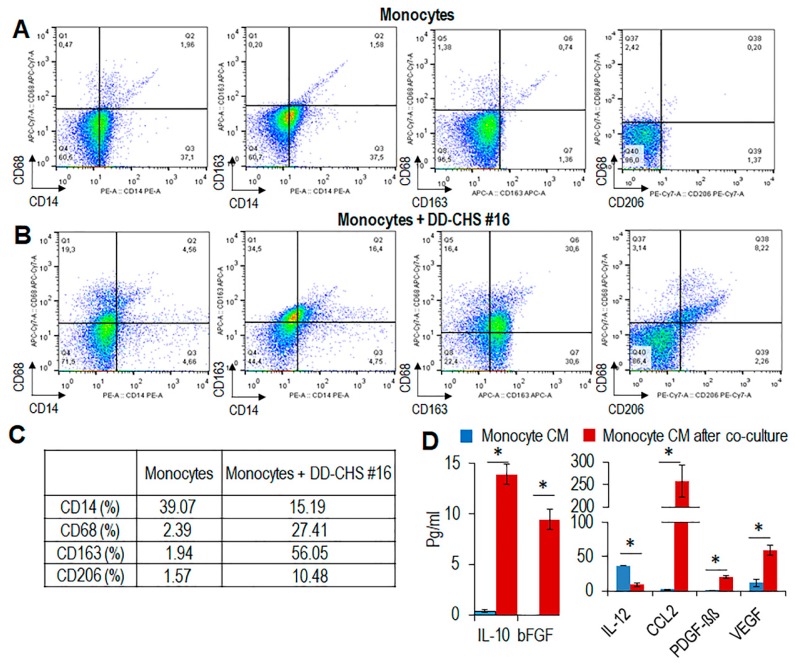
Immunophenotyping of primary human monocytes co-cultured with CHS cells. Human monocytes from healthy donors were co-cultured with primary CHS cells. (**A**,**B**). After 72 h, monocytes were recovered and their phenotype analyzed by flow cytometry. (**C**). Percent variation of CD14^+^, CD68^+^ CD163^+^, and CD206^+^ cells upon co-culture with CHS cells, compared to control (monocytes alone). (**D**). After co-culture, CHS were removed, and monocyte conditioned media were prepared and analyzed by a Bio-Plex immunoassay. The concentration of significant soluble factors (expressed in pg/mL) are reported as mean ± SD from two experiments performed in quadruplicate. Statistical significance with * *p* < 0.001.

**Figure 7 cells-09-01062-f007:**
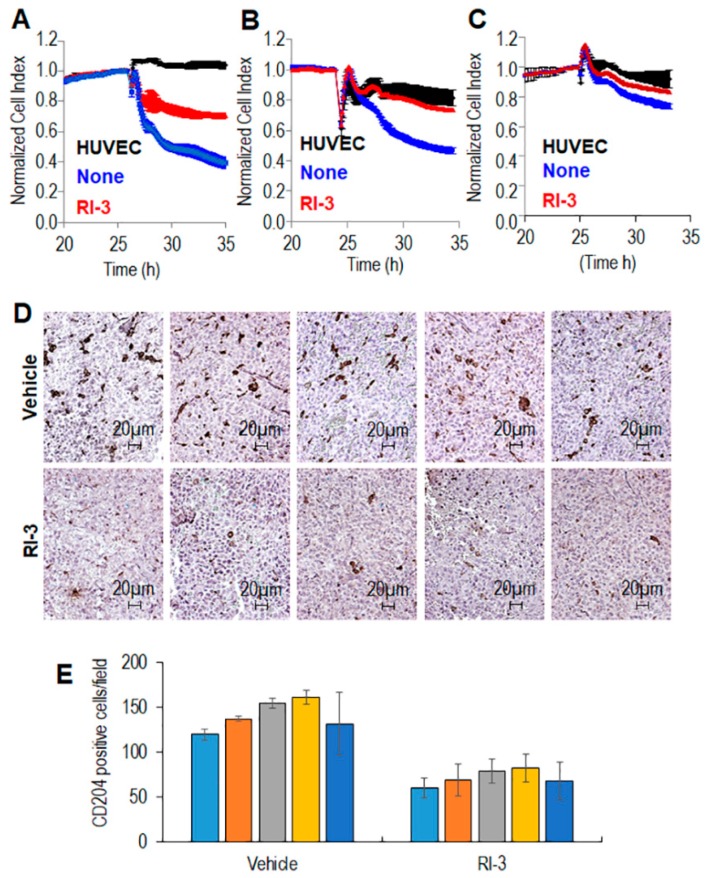
Effect of the peptide RI-3 on monocyte recruitment and TAM infiltration into CHS tissues. Endothelial cells (HUVEC) were seeded in E-plates and allowed to adhere for ~25 h, until they formed a confluent monolayer. Then, monolayers were overlaid with THP-1 cells (**A**), human (**B**) or murine (**C**) monocytes in growth medium, in the absence or presence of 10 nM RI-3. Endothelial invasion by monocytes was monitored in real-time for 10 h as changes in Cell Index. Values were normalized immediately after monocyte addition. Data represent mean ± SD from a quadruplicate experiment. (**D**,**E**). Ten 6–8 week-old Foxn1^nu/nu^ mice received an injection of CHS cells into the right flank as a single-cell suspension. Animals were randomized into two 5-mice groups with the treatment group receiving 6 mg/kg RI-3 by intra-peritoneal injection every 24 h, and the control group receiving an equivalent injected volume of vehicle. After 12 days, the animals were sacrificed, the excised tumors fixed in buffered formalin, processed for paraffin sectioning and then immunostained with anti-mouse-CD204. (**D**). Representative images of CD204 immunostaining are shown. (**E**). CD204^+^ cells revealed by IHC were counted in 5 randomly chosen fields per section, at 200× magnification and averages plotted. Statistical significance with * *p* < 0.0001.

**Table 1 cells-09-01062-t001:** Clinicopathological and Histopathological findings of enrolled chondrosarcoma patients.

Patients	Age (yr)	Gender	Site	Size (cm)	Histology ^a^	Grade	PFS ^b^
1	69	M	Sternum	10 × 7 × 5	CHS	G3	6
2	58	M	Sternum	18 × 15 × 8	CHS	G2	3
3	64	M	Left shoulder	20 × 16 × 15	CHS	G2	1
4	34	F	Left obturator ring	22 × 17 × 20	CHS	G3	51
5	67	F	Left distal femur	>5	CHS	G2	60
6	61	M	Left obturator ring	10 × 9 × 11	CHS	G2	ND
7	39	F	Right shoulder	5 × 4 × 3	CHS	G2	ND
8	75	M	Left proximal femur	13 × 8 × 11	CHS	G3	ND
9	62	M	Left distal femur	12	CHS	G2	ND
10	41	M	Left knee point	7 × 6 × 4	CHS	G2	ND
11	48	M	Left humerus	4.5 × 3.5 × 1	CHS	G2	ND
12	79	M	Left hand	3 × 5.5 × 4	CHS	G2	ND
13	72	M	Left proximal femur	21 × 12 × 12	DD-CHS	G3	3
14	63	F	Right iliac wing	13 × 9 × 13	DD-CHS	G3	14
15	77	F	Left proximal femur	9	DD-CHS	G3	10
16	64	F	Left humerus	10	DD-CHS	G3	ND
17	38	M	Left knee point	23	DD-CHS	G3	ND
18	71	F	Left humerus	14	DD-CHS	G3	ND

CHS, chondrosarcoma; DD, dedifferentiated; ^a^ Histopathological diagnosis was performed in according to the WHO 2013 classification; ^b^ Progression free Survival (months); F, female; M, male; ND, Not Determined.

**Table 2 cells-09-01062-t002:** CD68, CD163 positive cells, and intratumor microvessel density in CHS tissues.

	CD68		CD163		iMVD	
	Average ^a^	Score ^c^	Average ^a^	Score ^c^	Average ^b^	Score ^c^
1	200	5	173.3	5	153.8	5
2	94	3	21.75	1	25.5	2
3	80	3	29	3	28.25	3
4	153.5	5	121.33	4	73.5	3
5	81	3	7.5	1	29	2
6	220	5	111.33	4	129	4
7	239.5	5	175	5	159.66	5
8	230	5	179	5	163.67	5
9	136	4	69.25	3	34.75	2
10	64	3	20	1	5.6.	1
11	50	2	7.33	1	22.5	2
12	365	5	134.33	4	140	4
13	129.67	4	39	2	20	1
14	122.5	4	63.8	3	25.25	2
15	42	2	17.2	1	66.75	3
16	191.7	5	101	4	77.4	3
17	85	3	17	1	27	2
18	359.5	5	175	5	154	5

^a^ Average of CD68 or CD163 positive cells for field, counted in 5 field/sample, at 200× magnification. ^b^ Average of CD31 positive intratumor microvessels (iMVD) for field, counted in at list 5 field/sample, at 200× magnification. ^c^ Score: 0–25: Score 1; 26–50: score 2; 51–100: score 3; 101–150: score 4; >150: score 5.

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
