# Peer review of "Inhibiting Monocyte Recruitment to Prevent the Pro-Tumoral Activity of Tumor-Associated Macrophages in Chondrosarcoma"

_cells, 2020, doi:10.3390/cells9041062_

Round 1

Reviewer 1 Report

In this study "Inhibiting monocyte recruitment to prevent pro-tumoral activity of tumor-associated macrophages in chondrosarcoma". The authors examined the effect of tumor-associated macrophages (TAM)s in CHS. The results indicated that 3D organotypic co-cultures were set up in order to evaluate the contribution of primary human CHS cells in driving an M2-like phenotype in monocyte-derived primary macrophages, and in turn, the capability of macrophages to promote growth and/or invasiveness of CHS cells. This study is well designed.

  1. The clinic application is important. The authors should discuss how this study apply to clinic use
  2. Fig. 3&D. and 4A. The scale bar should be add as reference.
  3. The authors used contorl and RI-3 peptide to examined the effect. However, the detail information about these peptides should be provided.
  4. Table 1. There are many patients are lack information in Grade stage. Is any information about metastasis in these patients ?

Author Response

Response to Reviewer 1 Comments

In this study "Inhibiting monocyte recruitment to prevent pro-tumoral activity of tumor-associated macrophages in chondrosarcoma". The authors examined the effect of tumor-associated macrophages (TAM)s in CHS. The results indicated that 3D organotypic co-cultures were set up in order to evaluate the contribution of primary human CHS cells in driving an M2-like phenotype in monocyte-derived primary macrophages, and in turn, the capability of macrophages to promote growth and/or invasiveness of CHS cells. This study is well designed.

Point 1: The clinic application is important. The authors should discuss how this study apply to clinic use.

Response 1: In the revised version of this manuscript, the possible clinic application of inhibitors of monocyte recruitment into chondrosarcoma have been included in the Discussion on pages 20-21  as follows: “Indeed, many efforts have been directed to the prevention of monocyte recruitment into tumor tissues, counteracting their M2-like polarization, or, alternatively, forcing their phenotype toward a M1 pro-inflammatory phenotype with new compounds now in preclinical and clinical trials [63, 64]. However, some of these strategies have important limitations due to their specificity and toxicity therefore need further investigation [64]. (60, Cassetta, and Pollard. cancer. Nat Rev Drug Discov 12018. However, some of these strategies have limitations due to their specificity and toxicity and need further investigation (Cassetta and Pollard. Nat Rev Drug Discov .(2018)........... RI-3 may be considered a promising lead for development of new therapeutic strategies aimed to counteract the pro-tumoral effects of TAMs in chondrosarcoma. This information, together with the potent anti-angiogenic activity of RI-3, encourage to consider this peptide as a promising lead compound for development of new therapeutic strategies aimed at counteracting CHS progression.”

Point 2: Fig. 3&D. and 4A. The scale bar should be add as reference.

Response 2: As required, scale bars have been included in the Figure 3 C-D and Figure 4 A.    

Point 3: The authors used control and RI-3 peptide to examined the effect. However, the detail information about these peptides should be provided.

Response 3: In the revised manuscript detailed information on the peptide RI-3 has been included in the Discussion as follows: “Previous work from this laboratory has shown that uPAR-derived synthetic peptides carrying the S90P or S90E amino-acidic substitutions, up- or down-regulate cell migration, respectively [35]. Following this observation, we developed a series of linear peptides containing substitution of Ser90 with the glutamic acid or α-aminoisobutyric acid residues in the uPAR sequence that inhibit the uPAR/ FPR1 interaction and reduce to basal levels directional cell migration [32, 66]. While providing proof-of-principle for the strategy, none of these peptides represents an ideal lead molecule: some of these peptides are unstable to enzymatic digestion in human serum, whereas others exhibit toxicity when administered in vivo, probably due to a low affinity binding site to the alpha chain of vitronectin receptor [32, Carriero et al. Mol Cancer Ther. 2014].  To generate more stable uPAR/FPR1 inhibitors, we applied the retro-inverso approach to our previously described uPAR/FPR1 inhibitors [19]. Among these, the retro-inverso peptide Ac-(D)-Tyr-(D)-Arg-Aib-(D)-Arg-NH2, named RI-3 is stable RI-3 is stable in human serum and has no effect on cell proliferation, even at a 10μM concentration [19-20]. At nanomolar concentrations, it inhibits migration, matrigel invasion and trans-endothelial migration of human sarcoma cells [19]. In nude mice engrafted with CHS cells, RI-3 caused a significant reduction of circulating tumor cells and intra-tumor vascularization, the latter being due to prevention of the VEGF-driven angiogenesis [19]”

Point 4: Table 1. There are many patients are lack information in Grade stage. Is any information about metastasis in these patients ?

Response 4: Grade stages of DD-CHS have been included in the Table 1. Regarding information about metastatic dissemination in patients, the following sentence has been included in the Results on page 8 : “Metastatic lesions occurred in five CHS patients (# 1, 2, 3, 4 10) and in two DD-CHS (#14, 16) patients. CHS patients # 2, 3 and 4 died a few months after surgery”.

Finally, typos have been corrected and the general style improved.

Reviewer 2 Report

This study “Inhibiting monocyte recruitment to prevent pro-2 tumoral activity of tumor-associated macrophages in 3 chondrosarcoma” is very interesting. Especially focus on the mechanism of monocyte recruitment. You can find my comment below:

  1. The patient number in Table 1 in the article is wrong, and the order 12 should be revised to 10.

  1. What is the basis for the presentation of the patient's number sequence in the article? It is somewhat complicated, and it is recommended to arrange them according to the two types of tumors: CHS and DD-CHS.

  1. In Figure 4, patient No. 9 and patient No. 14 are selected as the comparison value. In addition to the difference between DDCHS and CHS, is there any reference standard, because the difference in gender and age is close to 10 years old, why not choose the difference? Compare less? Are there differences in other patients?

  1. Although A and B in Figure 1 are both CHS, the overall expression levels of CD163 and CD68 are close to the DD-CHS specimen (Fig.1 B-D). Unlike the text, there is a significant difference between CD68 and CD163. More significant pictures can be exchange.

  1. This study provides a lot of strong evidence support for CHS for TAM caused by macrophage, but Figure 7 only proves that RI-3 can effectively reduce the recruitment of angiogenesis and mononuclear balls, but it should be directly linked to the inhibition of uPAR to reduce TAM. The evidence is a little weak, can you provide some more evidence or explain clearly.

  1. The TAM affected by M2-like Macrophage has a wide range of coverage, including angiogenesis, metastasis, and EMT. The popular immunosuppressive drug pathway PD-1 is also among them. I do n’t know whether it is possible to observe the effect of uPAR inhibition on other capabilities. , Can further prove the significant benefit of uPAR inhibition on the entire TAM.

Author Response

This study “Inhibiting monocyte recruitment to prevent pro-2 tumoral activity of tumor-associated macrophages in 3 chondrosarcoma” is very interesting. Especially focus on the mechanism of monocyte recruitment. You can find my comment below:

Point 1: The patient number in Table 1 in the article is wrong, and the order 12 should be revised to 10.

Response 1: We apologize for the mistake which is now corrected       

Point 2: What is the basis for the presentation of the patient's number sequence in the article? It is somewhat complicated, and it is recommended to arrange them according to the two types of tumors: CHS and DD-CHS.

Response 2: In truth, we organized the Table 1 by listing patients based on their enrollment data. However, we thank the Reviewer for the suggestion as we realized that, as it is, Table 1 is confusing. In the revised Manuscript, Table 1 has been completely reorganized according to the two types of tumors: CHS and DD-CHS. Consequently, also Table 2 has been rearranged and patient numbers have been modified in the entire manuscript.

Point 3: In Figure 4, patient No. 9 and patient No. 14 are selected as the comparison value. In addition to the difference between DDCHS and CHS, is there any reference standard, because the difference in gender and age is close to 10 years old, why not choose the difference? Compare less? Are there differences in other patients?

Response 3: At the beginning of this project, we were perfectly aware of that obtaining primary cultures from fresh tissues is a difficult task, especially when cartilagineous nodules are present. Furthermore, as already documented, it is very hardly to recover, enough primary tumor cells from CHS tissues, in order to perform the programmed experiments.  Thus, we have not been able to analyze possible differences between gender age or grade in other patients. The aim of experiments shown in the Figures 3 and 4, was limited to the analysis of pro-invasive ability of monocytes in co-cultures with CHS and DD-CHS primary cells in multicellular organotypic assays.                                         

Point 4: Although A and B in Figure 1 are both CHS, the overall expression levels of CD163 and CD68 are close to the DD-CHS specimen (Fig.1 B-D). Unlike the text, there is a significant difference between CD68 and CD163. More significant pictures can be exchange.

Response 4 We confirm that images of CD163  from the CHS patient #8 (ex  #14) and  DD-CHS patient #16 (ex #9), are shown in the panels  A and B, respectively, of Figure 1. We agree with the Reviewer that in the panel A the image showing the CD163+ cells does not reflect the average number of cells counted in in five randomly selected fields/sample at 200× magnification. Now, this image has been substituted with a more representative one.

Point 5: This study provides a lot of strong evidence support for CHS for TAM caused by macrophage, but Figure 7 only proves that RI-3 can effectively reduce the recruitment of angiogenesis and mononuclear balls, but it should be directly linked to the inhibition of uPAR to reduce TAM. The evidence is a little weak, can you provide some more evidence or explain clearly.

Response 5: In the revised manuscript,  Discussion has been reorganized and now includes new paragraphs in which more information on uPAR and RI-3 has been included as follows: “We and others have documented that the expression of urokinase receptor (uPAR) in monocytes increases with differentiation to macrophages and contributes to their activation, thus regulating immune responses, inflammation, and tumor progression [24, 25]. Also, in a variety of cancer cells, uPAR expression has been shown to induce, secretion of IL-4, via a mechanism that involves activation of ERK1/2, which, in turn, promotes M2 polarization of macrophages [65]. On the other hand, we have already documented that: i) uPAR is highly expressed in human CHS tissues [31]; ìì) CHS tumor cells expressing uPAR and the formyl-peptide receptor type 1 (FPR1) acquire the ability to migrate and invade basal membranes [19]; ììì) uPAR triggers cell migration through the interaction of its 84-95 sequence with the FPR1 [35]. Previous work from this laboratory has shown that uPAR-derived synthetic peptides carrying the S90P or S90E amino-acidic substitutions, up- or down-regulate cell migration, respectively [35]. Following this observation, we developed a series of linear peptides containing substitution of Ser90 with the glutamic acid or α-aminoisobutyric acid residues in the uPAR sequence that inhibit the uPAR/ FPR1 interaction and reduce to basal levels directional cell migration [32, 66]. While providing proof-of-principle for the strategy, none of these peptides represents an ideal lead molecule: some of these peptides are unstable to enzymatic digestion in human serum, whereas others exhibit toxicity when administered in vivo, probably due to a low affinity binding site to the alpha chain of vitronectin receptor [32, 66].  To generate more stable uPAR/FPR1 inhibitors, we applied the retro-inverso approach to our previously described uPAR/FPR1 inhibitors [19]. Among these, the retro-inverso peptide Ac-(D)-Tyr-(D)-Arg-Aib-(D)-Arg-NH2, named RI-3 is stable RI-3 is stable in human serum and has no effect on cell proliferation, even at a 10μM concentration [19-20]. At nanomolar concentrations, it inhibits migration, matrigel invasion and trans-endothelial migration of human sarcoma cells [19]. In nude mice engrafted with CHS cells, RI-3 caused a significant reduction of circulating tumor cells and intra-tumor vascularization, the latter being due to prevention of the VEGF-driven angiogenesis [19]…………. RI-3 may be considered a promising lead for development of new therapeutic strategies aimed to counteract the pro-tumoral effects of TAMs in chondrosarcoma. This information, together with the potent anti-angiogenic activity of RI-3, encourage to consider this peptide as a promising lead compound for development of new therapeutic strategies aimed at counteracting CHS progression.

Point 6: The TAM affected by M2-like Macrophage has a wide range of coverage, including angiogenesis, metastasis, and EMT. The popular immunosuppressive drug pathway PD-1 is also among them. I do n’t know whether it is possible to observe the effect of uPAR inhibition on other capabilities. Can further prove the significant benefit of uPAR inhibition on the entire TAM.

Response 6: These questions  have  been discussed in the Discussion as follows: “Despite several preclinical studies and clinical trials aimed to identify druggable targets that may improve the prognosis of CHS patients, this bone sarcoma remains an orphan disease, mainly  because all studies are performed with small numbers of patients [2-4]. Furthermore, no effective treatments exist for advanced CHSs due to their resistance to radiotherapy and chemotherapy. In recent years, the growing interest in cancer immunotherapy reached the sarcoma field and a number of molecular profiling studies led to the identification of immune therapeutic targets in bone sarcomas (Heymann MF et al., Br J Pharmacol. 2020;10.1111/bph.14999. doi:10.1111/bph.14999 PD1 expression seems to have prognostic and therapeutic implications in CHS, whereas PD-L1and T-cell infiltrate were found highly expressed in dedifferentiated CHS (Kostine, M., et al., Modern Pathology, 29(9), 1028–1037. https://doi.org/10.1038/ modpathol.2016). Unfortunately, to date, the clinical responses in trials remain unsatisfactory, but highlight the need to better characterize the CHS microenvironment in an effort to improve the immunotherapeutic response. Unfortunately, he clinical responses in trials remain unsatisfactory, suggesting the need for better characterize the CHS microenvironment in an effort to improve the immunotherapeutic response. Emerging evidence suggests that tumor-associated macrophages (TAM)s promote tumor progression exerting immunosuppressive and pro-angiogenic activities [50–54], In CHS, TAMs constitute the main immune population  (Richert I. et al., Journal of Bone Oncology, Volume 20, 2020,https://doi.org/10.1016/j.jbo.2019.100271)”.          

Regarding the contribution to TAM depletion or reprogramming by uPAR inhibition, this point has been discussed as follows: “We and others have documented that the expression of urokinase receptor (uPAR) in monocytes increases with differentiation to macrophages and contributes to their activation, thus regulating immune responses, inflammation, and tumor progression [24, 25]. Also, in a variety of cancer cells, uPAR expression has been shown to induce, secretion of IL-4, via a mechanism that involves activation of ERK1/2, which, in turn, promotes M2 polarization of macrophages (Jingjing Hu et al. uPAR Induces Expression of Transforming Growth Factor β and Interleukin-4 in Cancer Cells to Promote Tumor-Permissive Conditioning of Macrophages. The American Journal of Pathology, Volume 184, Issue 12, 3384 - 3393). On the other hand, we have already documented that: i) uPAR is highly expressed in human CHS tissues [31]; ìì) CHS tumor cells expressing uPAR and the formyl-peptide receptor type 1 (FPR1) acquire the ability to migrate and invade basal membranes [19]; ììì) uPAR triggers cell migration through the interaction of its 84-95 sequence with the FPR1 [35]. Previous work from this laboratory has shown that uPAR-derived synthetic peptides carrying the S90P or S90E amino-acidic substitutions, up- or down-regulate cell migration, respectively [35]. Following this observation, we developed a series of linear peptides containing substitution of Ser90 with the glutamic acid or α-aminoisobutyric acid residues in the uPAR sequence that inhibit the uPAR/ FPR1 interaction and reduce to basal levels directional cell migration [32, 66]. While providing proof-of-principle for the strategy, none of these peptides represents an ideal lead molecule: some of these peptides are unstable to enzymatic digestion in human serum, whereas others exhibit toxicity when administered in vivo, probably due to a low affinity binding site to the alpha chain of vitronectin receptor [32, 66].  To generate more stable uPAR/FPR1 inhibitors, we applied the retro-inverso approach to our previously described uPAR/FPR1 inhibitors [19]. Among these, the retro-inverso peptide Ac-(D)-Tyr-(D)-Arg-Aib-(D)-Arg-NH2, named RI-3 is stable RI-3 is stable in human serum and has no effect on cell proliferation, even at a 10μM concentration [19-20]. At nanomolar concentrations, it inhibits migration, matrigel invasion and trans-endothelial migration of human sarcoma cells [19]. In nude mice engrafted with CHS cells, RI-3 caused a significant reduction of circulating tumor cells and intra-tumor vascularization, the latter being due to prevention of the VEGF-driven angiogenesis [19]……… RI-3 may be considered a promising lead for development of new therapeutic strategies aimed to counteract the pro-tumoral effects of TAMs in chondrosarcoma. This information, together with the potent anti-angiogenic activity of RI-3, encourage to consider this peptide as a promising lead compound for development of new therapeutic strategies aimed at counteracting CHS progression.

Reviewer 3 Report

The authors provide sound data to show that in chondrosarcoma similar mechanisms between cancer cells and macrophages can be demonstrated as has been described in other types of solid cancers and further, compounds inhibiting monocyte/macrophage migration may be an interesting option for further drug development.  The manuscript is well written and the data presented justifies the conclusions. There are only a couple of minor things (typos, style) that could be addressed prior publication. These can be inspected by editorial bases without reviewer's further involvement.

Major problems: -

Minor issues:

Lane 39 Immunohistochemistry -> immunohistochemistry

Lane 60. Chodrosarcoma is pretty rare, so from the point of view of national health care it is hard to see as a huge clinical challenge (but may be from the doctor's). I would be better describe it as a difficult challenge

Table 1. Hystology -> Histology

Author Response

The authors provide sound data to show that in chondrosarcoma similar mechanisms between cancer cells and macrophages can be demonstrated as has been described in other types of solid cancers and further, compounds inhibiting monocyte/macrophage migration may be an interesting option for further drug development.  The manuscript is well written and the data presented justifies the conclusions. There are only a couple of minor things (typos, style) that could be addressed prior publication. These can be inspected by editorial bases without reviewer's further involvement.

Major problems: -

Minor issues:

Point 1: Lane 39 Immunohistochemistry -> immunohistochemistry

Response 1: This  has been corrected.

Point 2: Lane 60. Chodrosarcoma is pretty rare, so from the point of view of national health care it is hard to see as a huge clinical challenge (but may be from the doctor's). I would be better describe it as a difficult challenge.

Response 2: We thank the Reviewer for this suggestion. In the revised Manuscript, the following paragraph has been included at the begin of the Discussion on page 20 as follows: “Chondrosarcomas (CHS) constitute a malignant group of rare cartilaginous matrix-producing neoplasms, with diverse morphological features and clinical behavior [1]. Patients diagnosed with chondrosarcomas (CHS) are subjected to large tumor resections to prevent local recurrences or metastasis, including amputations, with physical disabilities that highly affect daily life. Despite several preclinical studies and clinical trials aimed to identify druggable targets that may improve the prognosis of CHS patients, this bone sarcoma remains an orphan disease, mainly  because all studies are performed with small numbers of patients [2-4]. Furthermore, no effective treatments exist for advanced CHSs due to their resistance to radiotherapy and chemotherapy.”

Point 3: Table 1. Hystology -> Histology

Response 3: This has been corrected

Finally, typos have been corrected and the style improved.

Round 2

Reviewer 1 Report

Accept to publish